# *Mycoplasma pneumoniae* Pleural Effusion in Adults

**DOI:** 10.3390/jcm11051281

**Published:** 2022-02-26

**Authors:** Chang Ho Kim, Jaehee Lee

**Affiliations:** Department of Internal Medicine, School of Medicine, Kyungpook National University, 680 Gukchaebosang-ro, Jung-gu, Daegu 700-842, Korea; kimch@knu.ac.kr

**Keywords:** *Mycoplasma pneumoniae*, pleural effusion, diagnosis, adult

## Abstract

Parapneumonic effusions often complicate *Mycoplasma pneumoniae* (MP) pneumonia, contrary to the notion that they are a rare feature of MP infection. Increased research and evidence on MP parapneumonic effusions (MPPE) can help elucidate its clinical significance as one of the variable manifestations of MP infection. This article aims to summarize the existing literature about the clinical characteristics of MPPE in adults and discuss its diagnostic implications from the perspective of pleural fluid analysis. Approximately 20–25% of adult patients with MP pneumonia develop MPPE, and its frequency in children and adults seems to be similar. Although the pathogenesis of MPPE remains to be elucidated, MP-induced cell-mediated immune mechanisms might be partially associated with the development of MPPE. MPPE usually shows mononuclear leukocyte predominance with elevated adenosine deaminase (ADA) activity, similar to tuberculous pleural effusion (TPE). The degree of increase in pleural fluid ADA levels and serum inflammatory biomarkers may help differentiate between MPPE and TPE. During the acute phase, a single positive IgM and positive polymerase chain reaction results allow for a precise and reliable MP infection diagnosis. The mainstay of treatment is the selection of adequate anti-mycoplasma antibiotics with or without corticosteroid, based on the local epidemiologic data on macrolide resistance.

## 1. Introduction

*Mycoplasma pneumoniae* (MP), which was first discovered in the 1940s, causes a wide spectrum of clinical symptoms and diseases [1,2]. Pleural effusions frequently complicate MP pneumonia similar to complications in pneumonia caused by other bacterial pathogens [3,4,5,6,7,8], which is contrary to the notion that MP parapneumonic effusions (MPPE) are so rare that they receive a different diagnosis than MP infections [9]. Based on the literature from the last few decades, approximately 20–25% of patients with MP pneumonia develop parapneumonic effusion [3,4,5,6,8]. The latest reports suggest that the clinical significance of MPPE needs to be emphasized in terms of differential diagnosis with tuberculous pleural effusions (TPE) [8,10,11,12,13,14]. A better understanding of the characteristics of MPPE will contribute to the accurate diagnosis and appropriate treatment of the condition. Thus, the present review focuses on the clinical and diagnostic implications of MPPE from the perspective of pleural fluid analysis in adults.

## 2. Epidemiology

MP infections occur both endemically and epidemically worldwide [15] and are estimated in the range of 2–12% among adults with community-acquired pneumonia [16,17,18]. The occurrence of MPPE has been reported in 4–28% of MP pneumonia cases [3,4,5,6,7,8,19,20,21]. This variable incidence of MPPE may be influenced by the radiologic modalities used to detect pleural effusions [3]. A prospective study using serial chest X-rays, including lateral decubitus views by Fine et al. [5] reported that pleural effusions were identified in 21% of patients with MP pneumonia. They suggested not dismissing the possibility of MP infections in patients with parapneumonic effusion. Another study reported that, among 54 adult patients with MP pneumonia, pleural effusions were detected in 24% of them using computed tomography (CT), while only a half of them could be detected using posteroanterior and lateral chest X-rays [3]. Similarly, a study performed during an epidemic of MP infection revealed that 28% of patients with MP pneumonia had pleural effusions based on chest CT images [8]. The tendency of increase in the frequency of MPPE may be due to the advanced diagnostic methods for MP infection and widespread use of CT modality. The reported incidences of MPPE in children and young adults were similar [4,5,6,7,8].

## 3. Pathogenesis

The mechanisms by which MP, the smallest self–replicating bacteria lacking a cell wall, causes infection are complex [15,22,23,24,25]. MP can cause direct damage to the host via adhesion, membrane fusion, nutrition depletion, invasion, and toxin-like effects [25]. In addition, MP induces collateral damage to host tissues by immune responses during MP infection, which include humoral and cellular immune responses, inflammatory and antigenic damage, and immunosuppression [25]. Adhesion to the host cell is the first step in its pathogenicity mechanism [15,24]. The adhesion protein complex, which consists of the key protein P1 and several auxiliary proteins, enables P1 proteins to adhere to host cells [24,25]. Once attached, MP produces a variety of substances that cause local damage to epithelial cells and stimulates the release of inflammatory mediators from host immune cells [22,23,24,25].

Parapneumonic effusions usually occur when the pathogens enter the pleural space directly by breaching the pleural barrier through the alveolar air space [26]. Pleural mesothelial cells, the first to encounter the invading pathogens, have surface molecules including Toll-like receptors, monosialyl containing receptors, and glycoproteins to serve as receptors for bacterial determinants [26,27]. MP is likely to first interact with mesothelial cells during pleural infection, similar to the way that it adheres to host respiratory epithelial cells during lung infection [24]. Previous reports on mesothelial cell proliferation in the pleural fluid of patients with MPPE [28,29] support this speculation. Subsequently, MP may induce host innate and acquired immune responses via enhanced expression of Toll-like receptors within the pleural space [15,23,24,25].

A characteristic feature of parapneumonic effusions is generally the accumulation of neutrophils and mononuclear phagocytes, which are the most important components of the initial innate immune response during bacterial infection [30,31,32]. This finding contrasts with previous studies on adult patients with MPPE, which report that lymphocytes are usually predominant in the acute stage (Table 1) [6,8,10,11,29,33,34,35,36,37,38,39,40]. Histopathological examination also revealed chronic inflammation and accumulation of numerous lymphocytes and mononuclear cells on the pleural tissue obtained from a patient with MPPE [11]. This pleural fluid feature of MPPE is consistent with what may be expected from an intracellular pathogen, such as *Mycobacterium tuberculosis* (MTB), or a virus. Lymphocytic exudate of TPE develops due to the proliferation of antigen-specific T cells under the cell-mediated immune response to MTB [41]. Even though MP is primarily an extracellular pathogen, cell-mediated immune mechanisms play an important role in the development of MP infection [15,22,23]. T–lymphocyte activation and different cytokine stimulation patterns have been reported to be related to the evolution of MPPE [42,43]. It remains to be elucidated whether the different cellular predominance in a pleural fluid depends on the degree of cell-mediated immune response and reflects the disease severity or phase of MPPE.

## 4. Clinical Manifestations

### 4.1. Clinical, Radiologic, and Laboratory Features

MPPE usually occurs as an acute illness with gradual onset. The most common presenting symptoms are high fever, nonproductive coughing, and pleuritic chest pain [6,8,11,29,33,34,35,36,37,38,39]. Other common symptoms include headache, malaise, and other extra-pulmonary phenomena [6,29,33,34,35,36,37,38,39]. MPPE mainly occurs in young adults (Table 1). The effusions are generally unilateral and small, and resolve with appropriate antimicrobial therapy [10,36]; however, they can be bilateral and large, and lead to shortness of breath occasionally [37,38,44]. The presence of pleural effusion in child and adult patients with MP pneumonia suggests a more severe clinical course, including higher levels of serum inflammatory biomarkers, increased extent of pneumonic infiltration, and prolonged hospitalization when compared to those without pleural effusion [4,8].

Effusions are always associated with pulmonary infiltrates, which are usually unilateral and located in the lower lobe [34,38,39,45]. Pleural effusions are more associated with alveolar or lobar pneumonia patterns than with interstitial pneumonia patterns in adults as well as children with MP pneumonia [3,46,47]. However, radiologic findings are nonspecific and are usually indistinguishable from other bacterial pneumonia patterns.

White blood cell (WBC) count is normal or mildly elevated. Serum C-reactive protein (CRP) and lactate dehydrogenase (LDH) are more elevated in patients with pleural effusion than in those without pleural effusion [4,10]. These findings suggest highly severe systemic inflammation as seen in other bacterial parapneumonic effusions. Other laboratory features are nonspecific.

### 4.2. Pleural Fluid Analysis

The MPPE is usually serous colored but may be purulent or bloody as other bacterial parapneumonic effusions [11,29,33,34,35]. The effusion is virtually always an exudate. Total leukocyte counts and routine biochemistries of MPPE are not different from the other bacterial parapneumonic effusions. The most distinct characteristics of MPPE are differential WBC counts and adenosine deaminase (ADA) levels.

#### 4.2.1. Differential WBC Counts

MPPE commonly shows mononuclear leukocyte (MN) predominance even during the acute stage, which is different from other bacterial parapneumonic effusions that exhibit polymorphonuclear leukocyte (PMN) predominance. In the data collected from reports of cases and small series providing differential leukocyte counts, MN was predominant in 68% of the cases (Table 1). Generally, the predominant cell population is determined by the cause and phase of pleural injury and the timing of thoracentesis in relation to the onset of pleural injury [45,48]. The acute response to most pleural injuries, whether infectious, immunologic, or malignant, is the attraction of neutrophils to the pleural space [49,50]. Thus, the PMN predominance usually suggests early-stage inflammation than MN predominance. However, shifting to MN predominance in MPPE seems to rapidly occur even during acute clinical illness compared to that in other common bacterial parapneumonic effusions. This phenomenon may be attributable to a particular feature of the MP pathogen mediating cellular immunity like intracellular pathogens, such as MTB, as stated in the section on pathogenesis.

#### 4.2.2. Adenosine Deaminase Levels

Roughly one in ten bacterial uncomplicated parapneumonic effusions, one-third of complicated parapneumonic effusions, and two-thirds of empyema have pleural fluid ADA levels above 40 U/L [51]. Despite neither complicated parapneumonic effusions nor empyema, MPPE seems to have a higher frequency of elevated ADA levels compared to that in other usual bacterial parapneumonic effusions. There was no data comparing pleural fluid ADA levels between MPPEs and other bacterial parapneumonic effusions. However, data from previous studies on adult patients with MPPE often reported higher levels of pleural fluid ADA; ADA levels of 7 out of 14 patients (50%) exceeded 40 U/L (Table 1). Additionally, elevated pleural fluid ADA levels are supported by the findings that serum ADA levels were higher in MP pneumonia cases than in other typical bacterial pneumonia cases [52,53,54]. These findings in conjunction with MN predominance led to confusion among the differential diagnosis between MPPE and TPE [11,14]. However, pleural fluid ADA levels of MPPE showed modest elevation like other bacterial parapneumonic effusions with elevated ADA activity, when compared to those of TPE. In contrast, systemic inflammatory biomarkers, such as serum CRP or LDH levels of MPPE were higher due to bacterial pneumonia compared to those of TPE [10,55]. These different properties may help distinguish MPPE from TPE.

## 5. Diagnosis

A definitive diagnosis of MP infection is a constantly challenging issue due to the fastidious nature of the pathogen, the considerable seroprevalence, and the possibility of transient asymptomatic carriage [56,57]. However, the early and precise diagnosis of MP infection and the use of appropriate anti-mycoplasma drugs are crucial factors to prevent the development of fulminant or fatal MP pneumonia [47,58,59]. Pleural fluid provides additional information and can be used for the diagnosis of MP infections. The current best algorithm to improve the sensitivity and specificity for acute MP infection diagnosis is a combination of polymerase chain reaction (PCR) and serology tests [56,57]. Although the isolation of MP from the pleural fluid, as well as sputum in patients with MPPE, has been reported [37,39,60], MP culture is time-consuming, requires special culture media, and has a high potential for false negatives, which is why it is not available in many centers [56].

### 5.1. Nucleic Acid Amplification Test

PCR tests as single or multiplex assays are the diagnostic method of choice for acute MP infection [56]. MP PCR using respiratory tract samples (nasopharyngeal swab, sputum, bronchial washing, bronchoalveolar lavage fluid, and pleural fluid and tissue) is fast and sensitive. However, it lacks specificity because it cannot distinguish between true patients and asymptomatic carriers [61]. Thus, positive results should be interpreted cautiously in context with clinical findings. Positive PCR results in the pleural fluid or tissue can establish a definite diagnosis [11,62]. Given the recent evidence that pleural biopsies improved microbiological yield in pleural infection [63], pleural tissue, only when available, may be a good sample for detection of MP, particularly through PCR tests.

### 5.2. Serologic Test

Enzyme immunoassay (EIA) is the preferred method for the detection of the specific anti-mycoplasma antibody and has largely replaced the complement fixation and microparticle agglutination tests [56,57]. EIA test also allows for the separate detection of IgM, IgA, or IgG. The gold standard of serologic diagnosis is a four-fold or greater increase of MP specific IgG when comparing acute and convalescent sera collected with an interval of 2–4 weeks. However, the need for a convalescent serum makes it impractical for the early management of acute illness. Using a single high IgM or IgA titer to make a presumptive diagnosis is an alternate strategy, although using only acute phase serology lacks both specificity and sensitivity [1].

### 5.3. Combination of Tests

There is currently no test that unequivocally distinguishes MP infection from a carriage or from a previous MP infection, except for positive PCR or culture result from pleural fluid or tissue. During the acute phase, a single positive IgM in combination with a positive PCR result allows for a precise and reliable MP infection diagnosis [64,65]. The clinician should attempt to incorporate patient characteristics, especially pleural fluid characteristics while considering the limitation described above [1].

### 5.4. Diagnostic Approach for Discriminating between MPPE and TPE

Recently, two reports of misdiagnosed MPPE and TPE were presented [11,14]. Because of the similar clinical course and characteristics of pleural effusion between MPPE and TPE, misdiagnosis is likely to occur in countries with intermediate or high prevalence of tuberculosis [11,13]. Figure 1 shows a diagnostic approach suggested for discrimination between MPPE and TPE. As with the investigation of any exudative pleural effusion, pleural fluid tests, including biochemistries, WBC counts, microbiological studies, and cytology are routinely performed [66]. In cases with an exudative pleural effusion of lymphocytic predominance without convincing evidence of etiology, routine pleural fluid ADA levels can be used as a predictive factor for TPE regardless of the prevalence of tuberculosis in the specific geographical area [66,67]. ADA levels below 40 U/L virtually rule out TPE and favor MPPE or other lymphocytic exudative pleural diseases. In contrast, pleural fluid ADA levels above 70 U/L suggest TPE rather than MPPE or others [10,68]. Alternatively, if pleural fluid ADA levels are between 40 and 70 U/L, many etiologies are possible. When the differential diagnosis is confined to MPPE and TPE, serum CRP levels exceeding 10 mg/dL that reflect more systemic inflammation usually accompanied with bacterial infection [69], are likely to favor MPPE, while those favoring TPE are usually less than 10 mg/dL [10,55]. Similarly, serum LDH levels reflect the inflammatory degree of MPPE, which is more elevated than those of TPE. Thus, the serum LDH/pleural fluid ADA ratio may be used as a parameter for distinguishing between MPPE and TPE [10,13]. In cases with an uncertain or atypical clinical picture, the final diagnosis can be established by pleural biopsy with tissue PCR and culture.

## 6. Management

Macrolide, tetracyclines, and respiratory fluoroquinolone are effective against MP. One of these agents is selected based on patient comorbidities, potential drug interactions, and the likelihood of macrolide resistance [70,71]. Due to the lack of a cell wall, MP is insensitive to ꞵ-lactam antibiotics. The optimal duration of therapy is not clear; however, 10–14 days or longer is generally recommended, according to clinical responses [15]. Pleural drainage can be considered in patients with worsening MPPE despite treatment with appropriate antimicrobials [36,39]. Corticosteroids with adequate antibiotics are a reasonable treatment option, especially in fulminant or refractory MP pneumonia patients that present with a hyperactive immune response [1,47,58].

The frequency of macrolide resistance has been increasing in various regions worldwide, particularly in Asia. In regions where the development of macrolide resistance is suspected based on local epidemiology research, such as most of Asia, alternative therapy (doxycycline or fluoroquinolone) should be considered in patients with severe or refractory diseases [71]. Macrolide resistance tests have been developed using a molecular method and are expected to be conducted more frequently [72].

## 7. Future Directions

Although there has been recent progress in the field of understanding MP infection, there remains a lack of high-quality data on MPPE. Therefore, a prospective investigation is warranted for a better understanding of this underexplored disease. In addition, further research is needed to explore how ADA, the levels of which are increased in various pleural effusions, acts differently and how we distinguish between these results.

## 8. Conclusions

Approximately one–fifth of adult patients with MP pneumonia develop MPPE. MPPE frequently shows MN predominance with modestly elevated ADA levels. These findings suggest a clinical implication of differential diagnosis with TPE. The degree of elevated pleural fluid ADA levels and serum inflammatory biomarkers may help differentiate between MPPE and TPE. The best current method to diagnose acute MP infection is a combination of PCR and serology. The mainstay of treatment is the selection of adequate antibiotics, such as macrolide, tetracycline, and respiratory fluoroquinolone, with or without corticosteroid, based on the local epidemiologic data for macrolide-resistance.

## Figures and Tables

**Figure 1 jcm-11-01281-f001:**
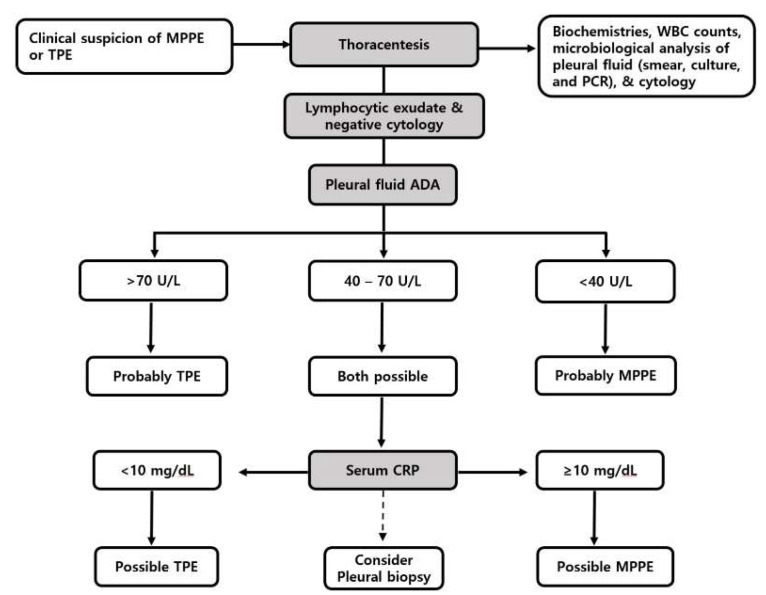
Proposed diagnostic algorithm for suspected patients with *Mycoplasma pneumoniae* parapneumonic effusion or tuberculous pleural effusion. MPPE = *Mycoplasma pneumoniae* parapneumonic effusion; TPE = tuberculous pleural effusion; WBC = white blood cell; PCR = polymerase chain reaction; ADA = adenosine deaminase; CRP = C-reactive protein.

**Table 1 jcm-11-01281-t001:** Literature review of pleural fluid analysis providing differential leukocyte counts or ADA levels in adult patients with *Mycoplasma pneumoniae* parapneumonic effusions.

No.	No. of Patients	Age (y)/Sex	Pleural Fluid						Year of Publication [Authors] [Reference No.]
			Amount	Total WBC (cells/μL)	Predominant Cell Type	Protein (g/dL)	LDH (U/L)	ADA (U/L)	
1	1	30/F	NA	NA	Ly	3.5	NA	NA	1971 [Feizi et al.] [29]
2	1	24/M	L	4800	Ly = Ne	5.4	NA	NA	1972 [Lewis et al.] [33]
3	1	23/M	L	650	Ly	3.6	NA	NA	1975 [Murray et al.] [37]
4	1	62/F	Mo	2600	Ly	1.8	104	NA	1975 [Chester et al.] [38]
5	1	21/F	L	4025	Ne	3.6	NA	NA	1976 [Gump et al.] [39]
6	1	18/M	L	41,000	Ne	4.0	835	NA	1979 [Solanki et al.] [34]
7	1	24/M	Mo	8350	Ne	NA	NA	NA	1984 [Linz et al.] [6]
	2	31/F	L	1320	Ne	NA	NA	NA	
8	1	37/F	S	10,000	Ne	5.4	469	19	2012 [Cha et al.] [8]
	2	18/F	S	500	Ly	5.5	632	88	
	3	20/F	S	475	Ly	2.3	330	43	
	4	24/F	Mo	5200	Ly	5.2	651	66	
	5	20/M	S	250	Ly	4.3	2320	51	
9	1	26/M	Mo	3540	Ne	3.5	452	18	2017 [Bajantri et al.] [35]
10 ^*^	1	76/F	S	1600	Ly	5.3	1058	28	2017 [Kim et al.] [10]
	2	30/F	S	425	Ne	2.5	244	10	
	3	42/F	S	1825	Ly	3.0	935	49	
	4	19/M	S	4502	Ly	3.8	1580	72	
	5	19/M	S	2145	Ly	3.8	948	31	
11	1	22/F	Mo	NA	Ly	NA	NA	NA	2018 [Hassan et al.] [36]
12	1	30/F	Mo	NA	Ly	4.8	685	46	2020 [Wen et al.] [11]
13 ^†^	1	34/F	NA	163	Ly	2.1	NA	11	2020 [Ding et al.] [40]
	2	32/M	NA	413	Ly	3.0	2322	25	
Total, median (IQR)/%	23	24 (20–32)/ F (61%)	S (45%)	1985 (481–4726)	Ly (68%)	3.7 (3.0–5.1)	685 (452–1058)	37 (19–55)/ ADA > 40 (50%)	

ADA = adenosine deaminase; WBC = white blood cell; LDH = lactate dehydrogenase; F = female; M = male; NA = not available; Ly = lymphocytic; Ne = neutrophilic; L = large; Mo = moderate; S = small; IQR = interquartile range. * Data for the remaining five patients were expressed after five patients included in the previous study (No. 8) were excluded. ^†^ Cases with severe M. pneumoniae pneumonia and acute respiratory distress syndrome.

## Data Availability

Not applicable.

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
