# Peer review of "Mycoplasma pneumoniae Pleural Effusion in Adults"

_jcm, 2022, doi:10.3390/jcm11051281_

Round 1

Reviewer 1 Report

Dear authors, thank you for a this article. I have found it well written, and informative. It is written so that it can inform general medical physicians as well as expert respiratory physicians. 

I would recommend this for publication. It would be good that the authors suggest a number of points for further research

  1. Propesctive research in those with mycoplasma pneumonia- radiology, serology etc so that good data is avaialable, rather than retrospective series
  2. Detection of mycoplasma in effusions- has this ever been done? 
  3. The distinction between ADA in the various effusions is interesting and one that perhaps merits further investigation. 
  4. There was some recent evidence from Rahman et al that suggested that in pleural infection, pleural biopsies are the best option to culture a pathogen- could they expand a bit further on the need for a pleural biopsy? 

I would review again. 

Author Response

Please find enclosed our response letter.

Reviewer 2 Report

The authors reviewed the parapneumonic effusion due to Mycoplasma pneumoniae pneumoniae in adults. Although the focus of this study is very interesting, the analyzed sample size of cases with pleural effusion is too small (Table 1). The authors should increase the cases with pleural effusion due to Mycoplasma pneumoniae pneumoniae in adults.    

Author Response

(The authors gave the same response as above.)

Reviewer 3 Report

In the pathogenicity section of Mycoplasma pneumoniae, it is better to discuss about adhesion protein and five direct damage mechanisms (adhesion damage, membrane fusion damage, nutrition depletion, invasive damage, toxic damage) and five types of immune damage, including humoral immune damage, cell immune damage, inflammatory damage, antigen immune damage and immunosuppression.

Author Response

(The authors gave the same response as above.)

Round 2

Reviewer 2 Report

I decided previously.